

# New Global Characterization of Landslide Exposure

Robert Emberson[1], Dalia Kirschbaum[1], Thomas Stanley[2]

[1] Hydrological Sciences Laboratory, NASA Goddard Space Flight Center, 8800 Greenbelt Road, Greenbelt, Maryland 20771
USA

[2] Universities Space Research Association/GESTAR, 7178 Columbia Gateway Dr, Columbia, Maryland 21046 USA

*Correspondence to*: Robert Emberson (robert.a.emberson@nasa.gov)

**Abstract.** Landslides triggered by intense rainfall are hazards that impact people and infrastructure across the world, but comprehensively quantifying exposure to these hazards remains challenging. Unlike earthquakes or flooding which cover large areas, landslides primarily occur in highly susceptible parts of a landscape affected by intense rainfall or seismic shaking, which may not intersect human settlement or infrastructure. Existing global landslide inventories generally include only those reported to have caused impacts, leading to significant biases toward both locations where impacts are common and areas with higher reporting capacity. To address the limits of report-based inventories, we have combined a globally homogenous landslide hazard proxy derived from satellite data with open-source datasets on population, roads and infrastructure to consistently estimate global exposure to landslide hazards. These exposure models compare favourably with existing datasets of rainfall-triggered landslide fatalities, while filling in major gaps in inventory-based estimates in parts of the world with lower reporting capacity. Our findings also, for the first time, distinguish relative levels of landslide hazard mitigation between different countries.

## 1. Introduction

Rainfall-induced landslides are an important natural hazard in many countries around the world, both as independent events and within larger chains of cascading hazards due to their role in downstream debris flow hazards. Current estimates of landslide impacts suggest that they cause thousands of fatalities annually (Froude & Petley, 2018; Petley, 2012) and billions of dollars of economic damage (Dilley et al., 2005). Global hazard estimates are an important way to understand the relative efficacy of hazard mitigation mechanisms between different countries, and also provide policy-makers with tools to estimate the future challenges associated with landslide hazards. However, few studies exist at present that provide a globally-consistent set of estimates for landslide hazard, and even fewer that attempt to characterize risk and exposure.

Most studies of landslide impacts rely on observations of specific landslide events and the associated reporting of the impacts. A small number of studies have estimated global economic impacts (Dilley et al., 2005; Guha-Sapir & CRED, 2019), while other important work has collated the fatalities associated with landsliding around the world to give crucial insight into impacts (Froude & Petley, 2018; D. Petley, 2012). The reliance of these studies on landslide inventories leaves them subject to known biases associated with these inventories. Specifically, there tends to be better reporting in developed countries (Kirschbaum et



al. 2010; Monsieurs et al., 2018) and a lack of public data about landslide occurrence and impacts in more remote regions, resulting in major blind spots in Africa, portions of the Andes, western China, and parts of Indonesia and the Philippines.

The global coverage of satellite data offers opportunities to fill in data gaps that result from inventory-based assessment of landslide hazards. NASA's Landslide Hazard Assessment for Situational Awareness (LHASA) model provides a near-real time near-global estimate of landslide hazard, based on a global susceptibility map and inputs from NASA precipitation estimates (Kirschbaum & Stanley, 2018). Using this model, it is possible to estimate relative changes in landslide hazard around the world each year. More importantly, this approach does not rely on local inventories to characterize the hazard, and therefore provides a near-global, consistent estimate of landslide hazard, encompassing the vast majority of populated areas. To address the need for globally consistent data on landslide hazard and exposure, we utilize an updated and enhanced version of the global susceptibility model defined by Stanley and Kirschbaum (2017) combined with a newly available 19 year IMERG rainfall product (Huffman et al. 2014) to estimate global landslide hazard, and then combine this with global estimates of population and critical infrastructure.

By leveraging satellite data and open-source information on infrastructure and population, we can for the first time understand the global distribution of landslide exposure and how it varies around the world in space and time and across socioeconomic variables such as population, critical infrastructure and road networks. This information can also be considered together with other datasets such as Froude and Petley (2018) to assess relative vulnerability to landslide exposure in different countries. A globally consistent model could support hazard mitigation decision making and planning, particularly in developing countries with limited reporting capacity; it would additionally allow for more consistent evaluation of landslide hazard going forward. The LHASA based outputs can provide a pixel-by-pixel assessment of hazard and exposure seasonally across the globe. This demonstrates the value of using remote sensing data in concert with ground-based inventories to provide a clear picture of the impacts associated with landslides around the world.

## 2. Methodology

To calculate global landslide hazard and exposure estimates, we have incorporated the average rate of hazard 'Nowcast' (a qualitative estimate of increased landslide hazard in a given location in near real time) issued by an updated version of the LHASA model, and combined them with openly available datasets of infrastructure at a 1km resolution across the world. These maps of exposure, both annual and as a monthly landslide climatology, are an important initial output in their own right, but we have further analysed the data to compare our outputs with existing estimates of global landslide hazard. This provides



key insights into where existing inventory biases may exist, as well as highlights which countries and regions are most exposed
to rainfall-triggered landslide hazard. Below, we detail the methods used to generate these outputs.

## 2.1. Nowcast climatology

The LHASA model is designed to provide near real time awareness of potential landslide activity through landslide 'Nowcasts'
(Kirschbaum and Stanley 2018). The algorithm uses a susceptibility map calculated from globally available estimates of slope,
lithology, forest cover change, distance to fault zones, and distance to road networks to provide a relative estimate of static
susceptibility (Stanley and Kirschbaum 2017). The susceptibility map is then compared with satellite-based precipitation
estimates from NASA's Tropical Rainfall Measuring Mission (TRMM) Multi-satellite Precipitation Analysis (TMPA) and
Global Precipitation Measurement (GPM) Integrated Multi-satellitE Retrievals for GPM (IMERG) rainfall product. To
characterize the potential for landslide triggering, an Antecedent Rainfall Index (ARI), or weighted accumulation from the last
seven days of rainfall, is calculated at each pixel. If the ARI value exceeds a threshold, either a moderate-hazard or a high-
hazard Nowcast may be generated if there is moderate to high susceptibility within that area. Nowcasts are issued at a 1km
pixel resolution every 3 hours.

We have updated the LHASA model for this study to incorporate data made available since the initial version of the model.
We term this revised model 'LHASA 1.1'. First, the global landslide susceptibility map (Stanley & Kirschbaum 2017) was
updated to include the 2018 data on forest loss (Hansen et al. 2018) and road density from the Global Roads Inventory Project
(Meijer et al. 2018). Previously, the forest loss data was upscaled as a binary variable representing either the presence or
absence of any 30m forest loss pixel within each 1km grid cell. However, this update represents forest loss at 1km as a fraction
of the 30m grid cells which have recently experienced forest loss. This change will de-emphasize the role of forest loss in
locations with little recent disturbance but will not change the effect on any 1km grid cell which has experienced total loss of
all forest cover. The susceptibility map was recomputed at 1km resolution using the same fuzzy overlay methodology as the
previous version.

Secondly, we have updated the rainfall input. Due to a recently released near 20-year record of IMERG (version 6B), we have
modified the precipitation inputs to LHASA in the following ways. First, we extend the LHASA model from 50 degrees N-S,
which was the latitudinal extent of TMPA, to the 60 degrees N-S extent of the IMERG product (Huffman et al. 2013). This
latitudinal expansion now includes most of Northern Europe and Canada, and the only populated areas excluded are in Northern
Russia, Iceland, some of Scandinavia and Canada. Because falling snow is an important component of precipitation at higher
latitudes but not a major trigger of landslides, we changed the precipitation variable considered from total precipitation to just





rainfall. The LHASA model does not consider snow avalanches. The effects of this change should be minimal in the tropical and temperate zones previously studied.

Finally, leveraging the new IMERG rainfall product we recompute the thresholds above which landslide activity is anticipated
at each pixel based on the 95th percentile of a 7-day ARI weighted rainfall accumulation. The model is then reprocessed from 2000-present, and we build a nearly 20-year record of landslide Nowcasts around the world. Averaging the Nowcasts by month, we construct a Nowcast climatology, or average landslide Nowcast rate for each pixel. We also compute annual Nowcast rates. This provides a globally consistent proxy for landslide hazard over the course of the year in each location. We term this as 'Nowcast density', and it represents a proxy for intensity of landslide activity. We can then combine this with data
on population and infrastructure to assess the relative exposure to landslides.

The result is a raster dataset at approximately 1km resolution for each month of the years in the IMERG record. We compute additional metrics such as the inter-annual variability in Nowcast frequency and standard deviations of Nowcast frequency. This information is incorporated into the annual exposure estimates to provide a measure of the variability. This uncertainty analysis is discussed in more detail below.

**2.2. Exposure datasets & integration with hazard**

We have overlaid the hazard footprints derived from the LHASA-based Nowcast climatology on top of publicly available datasets of population and infrastructure globally to map the exposure of these elements to landslide hazard. We have additionally aggregated these data at a national scale to compare with existing studies. Below, we first describe the datasets used, and then the approach taken to combine them with the hazard outputs.

We use population data from the Gridded Population of the World version 4 dataset (Doxsey-Whitfield et al., 2015), adjusted to the UN WPP Population Density for 2015. Use of this dataset is in line with other studies of population exposure to global hazards (Carrao, Naumann, & Barbosa, 2016; Dilley et al., 2005; Kleinen & Petschel-Held, 2007). The resolution of this dataset is the same as the LHASA Nowcast output – approximately 1km – and thus can be directly mapped onto the hazard data.

The definition of critical infrastructure can differ depending on the relevant stakeholder or location. The UN Global Assessment Report 2015 incorporates schools, hospitals and residential areas (De Bono & Chatenoux, 2014), and we use this as an initial basis for our estimates. We incorporate roads as defined in the Global Roads Inventory Project (Meijer et al., 2018), and amenities including hospitals, schools, fuel stations and power facilities as defined by OpenStreetMap. Both catalogs have a global extent and are updated regularly. Additionally, they offer a consistent set of data that can be compared



across the world. While there are some caveats to this comparison, which are discussed below, we suggest that these two
datasets are likely the best datasets with global coverage, open access, and recent updates.

The GRIP roads dataset harmonises nearly 60 datasets describing road infrastructure into a single, consistent dataset covering
222 countries (Meijer et al. 2018). GRIP incorporates roads derived from OSM as well as other data sources, and is considered
to be a harmonised global road catalog. The daily updates for OSM are not incorporated into GRIP, but we consider the

globally harmonised nature to be more important than a frequently updated catalog for the purposes of our study. This dataset
is a shapefile of linear features, which is not initially directly compatible with the 1km resolution landslide hazard outputs. To
connect the linear road dataset with the pixel-based Nowcast density data, we have used the Line Density tool in ArcGIS to
calculate the density of roads at 1km resolution with an output of a road density map with units of km/km2. Although the GRIP
database classifies roads in one of five classes depending on size and importance (e.g. primary highway, residential road), we

have not distinguished between these classes in our analysis. While economic impacts vary based on the type of road, our
analysis is meant to highlight the total potential exposed length for all types of roads.

OpenStreetMap (OSM) is a continually updated global map of infrastructure, roads, settlement and land uses (OpenStreetMap
contributors 2015). The updates are contributed by members of the public and the data is openly available for access in
shapefile and XML format. While differing levels of input from different parts of the world mean that there can be differences

in the level of completeness of the map depending on the region (Barrington-Leigh and Millard-Ball 2017), the specificity of
the data makes it an excellent source for infrastructure information. There is detailed classification of different features in the
map that allow us to isolate specific types of infrastructure, such as medical amenities or power stations. In addition, the open-
source nature of OSM means this approach is highly replicable. We have used the OSM Planet data file (a single XML
document of approximately 1TB, containing the information for every mapped feature in the OSM map) and parsed the xml

data using a Python-based script to obtain the density of critical amenities at a 1km resolution. We define critical amenities as
those labelled 'School', 'Hospital' 'Fuel Station', 'Power Station' and other 'Power' nodes (including substations and
transformers), based on the OSM feature definitions. The OSM Planet file was downloaded on June 24th 2019. The script used
to parse this file is available in the supplementary material.

To combine the roads datasets and OSM-derived critical infrastructure with the hazard outputs, we have multiplied each by

the Nowcast density for each full year in the IMERG archive (2000-2018) and taken the mean value and standard deviation.
The resulting datasets on exposure for population, roads, and critical infrastructure are all calculated at approximately 1km
resolution. We have also generated month-by-month exposure rasters to estimate the climatology of exposure for the same




exposed elements. Since these outputs are based upon the LHASA Nowcast output, it is important to clarify the units in which our estimates of exposure are expressed. Table 1 provides a summary of the units and the terms used in the study.

In Table 1, the units for each of the exposure outputs is also explained. We use the shorthand $Pop_{exp}$, $Road_{exp}$, and $Infr_{exp}$ to denote population, road and infrastructure exposure, respectively.

## 2.3. Error assessment

Kirschbaum and Stanley (2018) assess errors in the LHASA 1.0 Nowcast hazard estimates by comparison with historical
landslide events recorded in both the NASA Global Landslide Catalog (Kirschbaum et al., 2010) and the dataset of fatal landslides generated by Petley et al. (2007). They find relatively low False Positive Rates (~1%) and moderate to good true positive rates (24-60% for moderate hazard Nowcasts). However, both the Global Landslide Catalog and the data of Petley et al. (2007) are not complete, meaning that the true and false negative rates are not easily quantified. More succinctly, since a complete dataset of landslide occurrence does not exist, it is challenging to calculate the accuracy associated with any
independent landslide hazard estimate. Quantifying the relationship between Nowcast density and landslide probability for a given area remains an important step for future research.

To explore the relative variability in landslide activity, we estimate the standard deviation in annual Nowcast density at each point, based on the near-20 year IMERG rainfall input. We then propagate the error into the estimates for exposure for population, roads and critical infrastructure. The raster data for the standard deviations in error are available in the supplemental
data.

Estimating errors associated with OpenStreetMap data can be challenging, since the data quality is determined by volunteers who contribute to the map database. Broadly, we suggest it is appropriate to consider two distinct sources of error; the location accuracy of the individual points and infrastructure, and the completeness of the inventory. As discussed by Mooney and coauthors (2010), a lack of ground data across the world makes it challenging to assess the positional accuracy. However, in
some locations, data can be compared with existing sources. In the UK, Haklay (2010) suggests that OSM data points offer positional accuracy comparable with the Ordinance Survey Maps (the government standard). For the purposes of our study, where the maximum resolution available for the landslide hazard data is 1km, this positional accuracy is in excess of the requirements. However, completeness of the map is more problematic.

Barrington-Leigh and Millard-Ball (2017) assess the relative completeness of the OSM roads data on a country-by-country
basis, finding that OSM data in many developed countries is near-complete, although this declines in some states with lower GDP. The completeness varies within individual countries, with the most complete mapping observed in the highest density cities as well as the most sparsely populated areas (reaching a low in moderately populated areas). We assume that the estimate of completeness presented by Barrington-Leigh and Millard Ball (2017) for roads is applicable to other infrastructure; we are not aware of other global estimates of OSM completeness for specific infrastructure categories, so while this assumption may
not fully hold we suggest it is more informative to use this completeness estimate than none at all. Applying this as an error





systematically across our analyses is challenging; we can normalize national-level OSM based measurements by the completeness measure of Barrington-Leigh and Millard-Ball (2017), but at a pixel level we present the exposure 'as is', since we have no a priori concept of how to apply completeness estimates at this scale. To effectively normalise the exposure data at a country level, we provide the completeness measure derived from Barrington-Leigh and Millard-Ball (2017) in
Supplementary Table 1. In the figures in supplementary material that show Infr$_{exp}$ aggregated at a national level, we normalise the exposed elements by the total number of critical infrastructure elements in each country, which serves to provide a useful intercomparison of the relative hazard, and does not require completeness metrics.

The GRIP roads database (Meijer et al. 2018) draws a significant part of the road inventory from OpenStreetMap, and so is subject to some of the same error constraints. In Europe, the roads are derived primarily from OSM, although completeness in
this part of the world is near-perfect (Barrington-Leigh and Millard-Ball 2017). GRIP also uses OSM data in China, where there is a dearth of other freely available datasets. As such, completeness estimates in China are difficult to accurately characterize, and we do not attempt to do so. Elsewhere, GRIP incorporates other road datasets to supplement OSM. These input datasets are limited to those with positional accuracy greater than 500m, which precludes significant positional errors that would affect our km-scale analysis. We are not aware of estimates of the completeness of the GRIP dataset; since it
integrates datasets from all over the world, external validation datasets of completeness are unlikely to exist comprehensively. As such, while we note that there may be parts of the world where coverage is incomplete, we do not have strong constraints on this.

## 3. Results

Our analyses provide a global set of observations of landslide exposure, in both raster format and tabulated by country. The source data is available in the supplementary material associated with this study.

Figure 1 shows the population exposure annually for each 1km pixel and Figure 2 shows the exposure of population, roads, and critical infrastructure at the same scale for a portion of Northern Italy and the Alps, to highlight the nature of the different datasets. As can be observed in Figure 2, population and roads are significantly more widely distributed than critical
infrastructure. Infrastructure is instead concentrated primarily in urban centers, although power distribution infrastructure follows similar transportation corridors to road networks. In other parts of the world, there are significant levels of exposure of critical infrastructure to landslide hazard. The co-location of power distribution and road network exposure highlights the potential for complex post-landslide damage and multi-sector impacts.

For each country we have tabulated the aggregated values for Pop$_{exp}$, Road$_{exp}$, and Infr$_{exp}$, average annual Nowcast density.
We also show the total population, total length of roads from GRIP, and total number of OSM critical infrastructure elements; this allows for calculation of the fraction of total that is exposed for each of these aspects. To normalize the number of Nowcasts for each country, we divide by area in square decimal degrees, rather than square kilometers; since the Nowcast data is output on a grid based on decimal degrees. The same aggregation approach could similarly be used at a sub-national level to assess



relative impacts in different administrative areas. These data can be found in Supplementary Table 1, where all data necessary
to replicate these results is available.

We also list the OSM completeness estimates from Barrington-Leigh and Millard-Ball (2017), the fatalities per country due to
non-seismic landslides assessed by Froude and Petley (2018), and the landslide-linked economic impacts assessed by Dilley
et al (2005). These datasets are, to our knowledge, the most current datasets that assess landslide impact in terms of economic
cost and fatalities globally, and provide valuable points of comparison for our results. Comparison of calculated $Pop_{exp}$ with
recorded fatalities is shown in Figure 5, and comparison of $Road_{exp}$ with economic impacts from Dilley et al (2005) in Figure
6.

## 4. Discussion

The most striking initial result of our study is that significantly larger proportions of the globe are exposed to rainfall-triggered
landslide hazards than are often considered. Inventory-based assessments (e.g. Dilley et al. 2005) do not show significant levels
of landslide hazard and exposure in sub-Saharan Africa or much of Asia and South America, while we find that many of these
countries have significant proportions of the population and infrastructure exposed. It is perhaps not surprising that exposure
to landslide hazard is elevated in the major mountain belts of the Andes and the Alpine-Himalayan Orogeny, but there are
other key hotspots that may be less well known. These areas include much of Japan, the Rwenzori mountains in Africa, Central
America and Mexico, and much of the Caribbean. We find specific hotspots for certain cities within or near mountain belts;
this is particularly evident at the edges of large conurbations that abut mountainous areas, such as Taipei, Rio de Janeiro and
the edges of Tokyo.

While the zones of densely packed critical infrastructure such as schools and hospitals are also in general associated with these
urban areas, the impact of landslides on linear infrastructure is more widespread. Roads and power transmission facilities often
follow similar linear corridors, and where those intersect areas of high landslide hazard the relative exposure can still be
important. The localised impact of a single landslide impacting a densely populated urban zone may be very high, with several
critical infrastructural elements impacted. However, the likelihood of a landslide occurring somewhere along lengthy road or
power transmission segments in regional-scale rainfall events is higher, and an interruption to linear infrastructure may impact
lifelines that are relevant in disaster response. Thus the localised and distributed impacts should be considered alongside one
another, we suggest that highlighting the most vulnerable corridors for power transmission and road traffic is an important
subject for future work.

To explore these results against independent datasets of landslide hazard and risk, we have aggregated the data at a country
level (Supplementary Table 1). We can then highlight those nations with the highest landslide impact both in absolute terms
(total exposed people and infrastructure) and as a proportion of the overall population or infrastructure in that country.
As might be expected, countries with the largest population have the highest overall population exposure, although exposure
in China exceeds that of India despite having a smaller population. Exposure of roads is also greatest in China and the United
States, which are both highly populated with good OSM coverage. These absolute values are important, but we suggest that



more insight can be gained by assessing the relative exposure of population and infrastructure in each country, as well as by comparing the different relative values between nations.

Intercomparison of different countries can highlight those nations where the impact of landslides is greatest, and can draw attention to smaller, less developed nations where landslide statistics from report-based inventories may be lacking.

Figure 3 plots $Pop_{exp}$, normalised by area for each country against the average Nowcast density in that country, with colors denoting the geographic region. Results indicate that hazard and exposure are generally well-correlated across different

countries; similar relationships exist for both road exposure and critical infrastructure (see supplementary material for figures). At the highest end of this scale – i.e. those with high x-axis values - are smaller countries where mountainous terrain makes up much if not all of the area: Monaco, Bhutan, Andorra, and several Caribbean States: St Vincent and the Grenadines, Dominica, Grenada and St Lucia. In terms of population exposure, many countries in Asia and Africa have higher population exposure for an equivalent level of Nowcast density, when compared to European and some Central American countries. This

results from the generally higher population of these states.

Figure 4 plots the relative fraction of the population impacted by landslides. The relatively lower values in some of the larger countries like the United States and Brazil suggests that while the overall population impact is high in highly populated states, the relative impact can be more concentrated in smaller countries.

Given the large degree of variability in annual Nowcast frequency, inventories of reported landslides may miss the average

landslide rate in smaller countries if catastrophic landslides do not coincide with the sampling period for the inventory. The bulk of reported landslide events occur in larger nations where statistical variability of landsliding is likely damped over larger areas like Nepal, Taiwan, China and Japan. While we find high normalised hazard estimates in many of those states, our analysis also highlights smaller nations where the relative impact of landslides may be more significant on longer timescales. Alongside the previously mentioned nations, we also find several smaller states with higher proportions of exposed population;

Montenegro, Bosnia and Herzegovina, and Macedonia are notable in the Balkan area in particular.

To test whether the Nowcast-exposure estimates are a useful predictor of landslide risk, we can compare them to existing datasets. In Figure 5, we plot the total exposure of population in each country (in units of person-Nowcasts per year) against the landslide fatality dataset assembled by Froude and Petley (2018). This dataset, collected from 2004-2016, consists of 4862

separate landslide events that resulted in fatalities, and is the most comprehensive dataset for landslides that have caused fatalities in the world. Figure 5 highlights that there is a relatively strong correlation, with countries in Asia, Central America and Africa generally exhibiting higher numbers of fatalities for a given population exposure than observations in Europe.

In Figure 6, we plot the total road exposure against a derived metric of GDP impact from Dilley et al. (2005) based on the EM-

DAT landslide dataset. The EM-DAT based assessment divides the globe into 2.5 degree squares and does not present absolute values of total economic loss, but instead a relative decile (1-10 with increasing risk) ranking of grid cells based upon the



calculated economic loss risks. While this metric is not quantitative of the economic risk, we suggest that it is possible to compare these relative loss rates against our results. As with the comparison between $Pop_{exp}$ and fatalities, we see a relatively strong correlation. However, it is clear that the EM-DAT dataset is incomplete; the complete absence of data on costs associated

with landslides in African countries limits how effectively we can compare this inventory with our model estimates. The absence of data does further highlight the value of our globally consistent approach.

Although there are countries without data in the EM-DAT derived database, it may be possible to derive these missing values based on the relationship between $Road_{exp}$ and the countries where EM-DAT data exists (points in Figure 6) – i.e., to capture

the y-axis values based on a known x-axis value. Extrapolation and validation of this relationship is beyond the scope of this current work, but we suggest is an important topic for future research.

In order to learn which factors control the relationships between exposure and impact in different countries, we can combine the inventory data with our estimates and compare it with other variables. In Figure 7, we plot the number of fatalities recorded in the dataset of Froude and Petley (2018) divided by $Pop_{exp}$. This is subdivided by continent. We suggest that fatalities divided

by exposure provides a proxy for the degree of hazard mitigation in a given country; lower values indicate that for a given level of population exposure, fewer fatalities are observed. We find high variability in each continent, although in general there are lower levels of fatalities per unit exposure in Europe when compared to Central America and the Caribbean, as well as South America. Germany and Hong Kong, highly developed countries, have proportionally low fatalities despite high levels of exposure, likely a result of extensive mitigation efforts.


At the other end of the spectrum, some less developed countries exhibit higher fatalities for a given exposure; Sierra Leone, Burkina Faso, Haiti, Suriname, Bangladesh, Dominica and the Philippines have a significantly higher level of fatalities per unit of exposure. Some key outliers (Qatar and Bahrain) have high fatality per unit exposure, but these nations have very low overall exposure (see Supplementary Table 1) meaning that even a small number of fatalities increases the y-axis value in

Figure 7 to a large degree. This analysis, while not comprehensive, may inform national-level landslide risk management and provide insight into relative vulnerability to a given level of exposure.

To explore whether the variability in fatalities divided by $Pop_{exp}$ seen in Figure 7 is related to the level of development in each country, we have compared fatalities / $Pop_{exp}$ with 2018 GDP values for each country (World Bank 2019) *A priori*, we would expect countries with greater GDP to be capable of mitigating hazard more effectively, and thus have fewer fatalities for a

given level of exposure. However, while there is a small average decline in fatalities for a given exposure as GDP increases (Figure 8), with some high GDP countries showing the lowest fatality values (notably Germany and Hong Kong) there is a significant degree of variability in this relationship, suggesting there is a more complex relationship.

We note that comparing the model-based estimates of exposure with the fatality inventory of Froude and Petley (2018) in this manner may lead to erroneous conclusions if not considered carefully. While it is likely that many, if not all of the fatal

landslides in developed countries are accurately recorded, this may not be the case in states where disaster management is less





advanced. As such the lack of strong relationship between fatalities per unit exposure and GDP per capita observed in Figure 8 may represent gaps in the data in countries with lower GDP per capita, and thus a systematic bias within this analysis. Phrased differently, there may still be a relationship between GDP and fatalities for a given exposure level, but this may be masked by a lower reporting capacity in less-developed nations.


While these results provide an independent estimate of landslide hazard and exposure across the globe that does not rely on a specific inventory, there are still assumptions and limitations that should be considered to put these results in appropriate context.

The most important caveat associated with this data is that Nowcasts do not represent a guarantee of a landslide. The LHASA
model Nowcasts (Kirschbaum and Stanley 2018) are issued when there is an increased likelihood of a rainfall-triggered landslide, meaning the estimates of exposure represent the relative likelihood of exposure to landslides, rather than the reported impacts. As such, Nowcast number is a proxy for landslide hazard, rather than a quantifiable landslide hazard. However, we suggest that this disadvantage is more than offset by the global homogeneity and comparability of the Nowcast output.

Additionally, since we do not have global data to quantify the vulnerability of settlements and infrastructure to landslide hazard,
we cannot quantify the risk and impacts associated with landslide hazard. For example, data on fatalities associated with landsliding (Froude & Petley, 2018; Petley, 2012) quantifies the impacts, and while we can express our outputs in terms of relative proportion of population exposed to hazard, the lack of vulnerability data in our study represents an unconstrained source of variability if we compare those two datasets. Moreover, since the Nowcast output does not capture information about the size of a potential landslide in a given area, there may be differences in the severity of the landslide events that occur
depending on local factors (e.g. topography).

We note that we do not identify specific hospitals or schools as exposed to landslides. The resolution of our analysis remains coarse for individual points, and identifying specific locations could lead to overconfidence in exposure estimates. We acknowledge the importance of downscaling exposure estimates to those points, and suggest it is another important future direction for landslide exposure estimation.

The resolution of the Nowcast data also presents challenges to the interpretation. While a Nowcast estimate for a 1km x 1km grid cell provides an estimate of the landslide hazard therein, it does not provide information about where exactly a landslide may occur. Since infrastructure and population are unlikely to be evenly distributed within a grid cell (and are likely to be located further from areas of highest landslide susceptibility if risk mitigation measures have been adopted), elements that we describe as 'exposed to landslide hazard' may never actually be so. Given the resolution of our input hazard data, we suggest
that it is challenging to provide a more finely resolved estimate. This does highlight the need for effective downscaling methods that can be applied to coarse resolution rainfall data to assess local landslide hazard. We hope to address this in future work. In addition, while our analysis covers rainfall-triggered landslides, both anthropogenic and seismic triggered slope failure significantly contribute to global landslide impact. We suggest future work should seek to homogenise these diverse triggering factors.





The value of a homogenous global dataset is highlighted when comparing the relative exposure of population to landslide hazard based on our estimates with the GDP cost associated with landslides derived from Dilley et al. (2005). The prior study is based upon the EM-DAT inventory of damaging landslides, but the complete absence of data for countries in sub-Saharan Africa (see Supplementary Table 1) contrasts strongly with our results, which suggest that there is a significant proportion of the population in many sub-Saharan African countries exposed to landslide hazard.

## 5. Conclusions

Through combining rainfall, topography and other satellite-derived data, we have developed a long-term estimate of landslide hazard across the globe, which we have utilised to estimate the exposure of population and infrastructure to rainfall induced landslides. These estimates are globally consistent, and compare favourably with existing global datasets. When used in conjunction with datasets of landslide fatalities we can provide a nuanced picture of where and when landslides are most impactful. Our data highlights a potential higher prevalence of landslide hazards than previously documented in in small, mountainous nations and islands; while the absolute numbers of fatalities may be smaller, these represent locations with extremely high hazard and exposure. Further work is necessary to both test these results in a range of settings, consider additional triggering factors such as earthquakes and human impact, as well as to explore how global estimates can be downscaled and compared to more local estimates.

## Acknowledgements

All material necessary to replicate these results can be found in the supplementary material. The authors have no conflicts of interest, financial or otherwise. D.K. and T.S. are supported by a NASA DISASTERS program grant 18-DISASTER18-0022. R.E. is supported by a NASA Postdoctoral Fellowship administered by Goddard Space Flight Center. All authors were involved in study conceptualisation and writing of the manuscript. R.E. and T.S. carried out modelling and analysis. Map data copyrighted OpenStreetMap contributors and available from https://www.openstreetmap.org.

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

| Parameter | Specific Unit | Descriptive term (shorthand used in this study) | Explanation |
|---|---|---|---|
| Population exposure | Person-Nowcasts. $yr^{-1}$. $km^{-2}$ | $Pop_{exp}$ | The exposure is estimated as number of Nowcasts per year in each square km multiplied by the population in that square km. |
| Road exposure | Nowcasts.km.$yr^{-1}$.$km^{-2}$ | $Road_{exp}$ | Sum of Nowcasts per square km multiplied by km of road within that square km. |
| Infrastructure exposure | Nowcasts.element.$yr^{-1}$.$km^{-2}$ | $Infr_{exp}$ | Includes the following critical infrastructure categories: hospitals, schools, fuel stations, power generation and transmission |

**Table 1: Summary of terms used to describe infrastructure and associated units.**

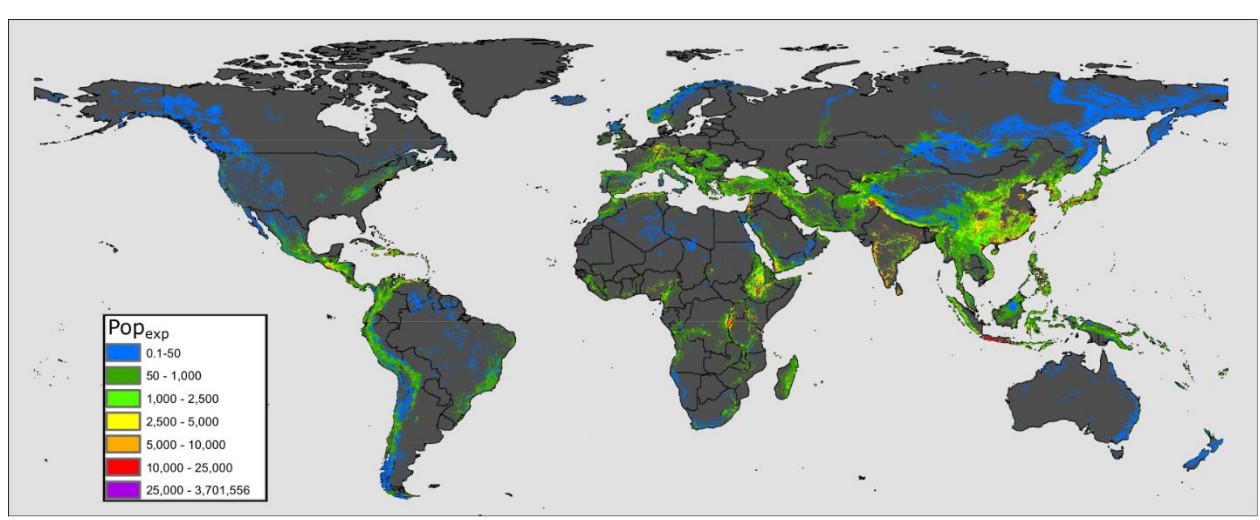

415

**Figure 1: Global population exposure to landslides ($Pop_{exp}$). Since the distribution of high-exposure areas is highly localised, we have binned the data to highlight differences at lower exposure levels more clearly. Country borders are drawn from the Natural Earth Dataset of Admin 0 boundaries (public domain).**





**Figure 2: Showing relative exposure of population, critical infrastructure, and roads in a snapshot of the world map - in this case, the European Alps and Italy. Country borders are drawn from the Natural Earth Dataset of Admin 0 boundaries (public domain).**

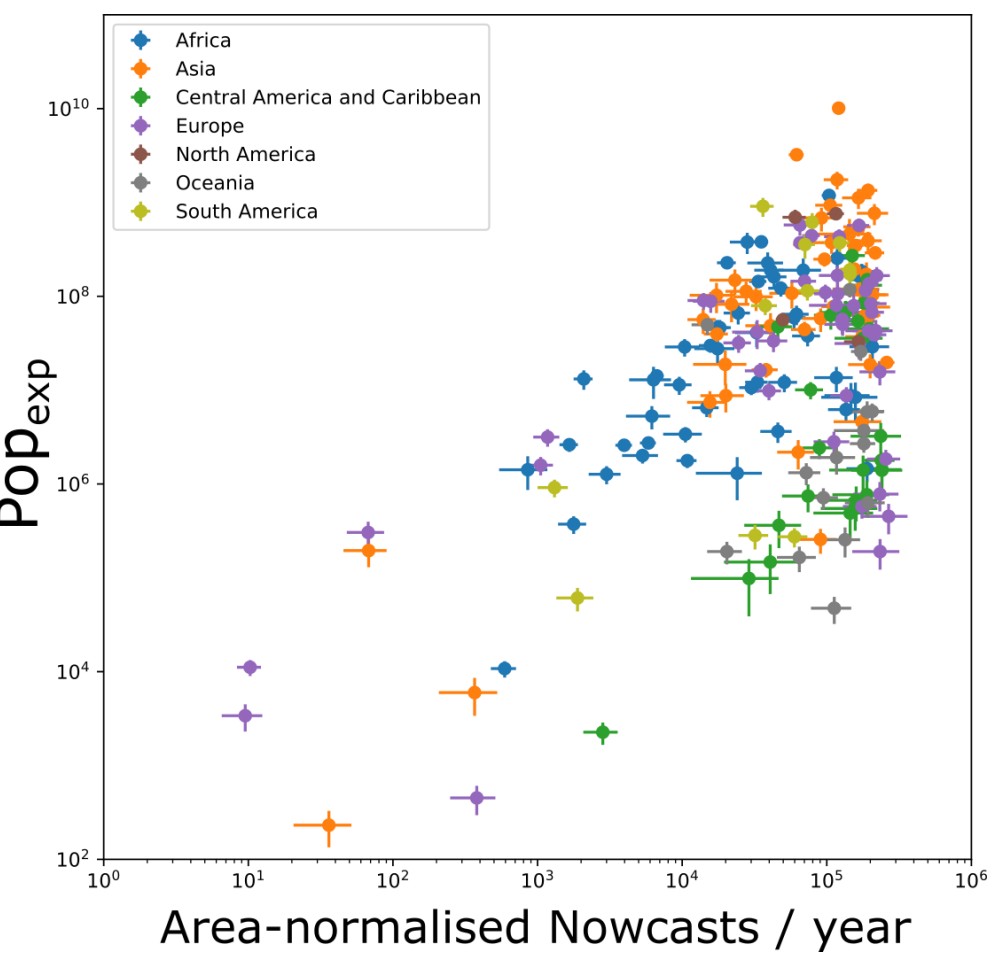

425

**Figure 3: Nowcasts per year, normalised by country area compared with the population exposed to Nowcasts (in units of Nowcast/person-years). Error bars show 1 standard deviation in inter-annual variability.**



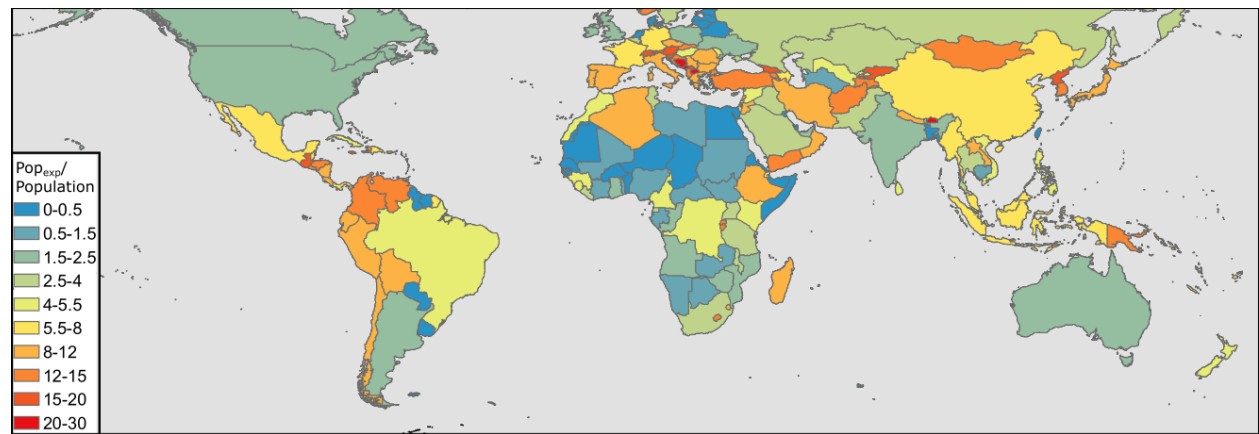

**Figure 4: Population exposure normalised by total population. This is expressed as Popexp divided by total 2018 population derived from the World Bank data archives (World Bank 2018). Country borders are drawn from the Natural Earth Dataset of Admin 0 boundaries (public domain).**

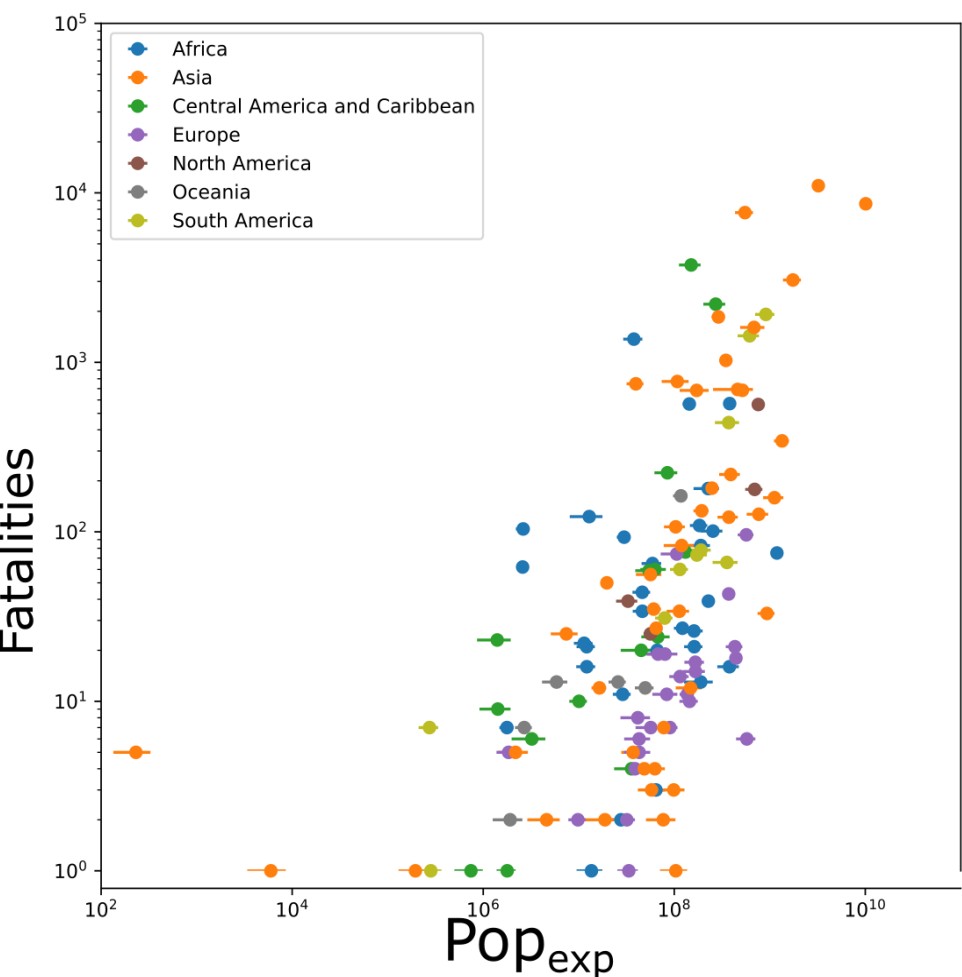

**Figure 5: Showing the exposure of population (in person-Nowcasts/year) against the number of fatalities recorded in the dataset of Froude and Petley (2018).**


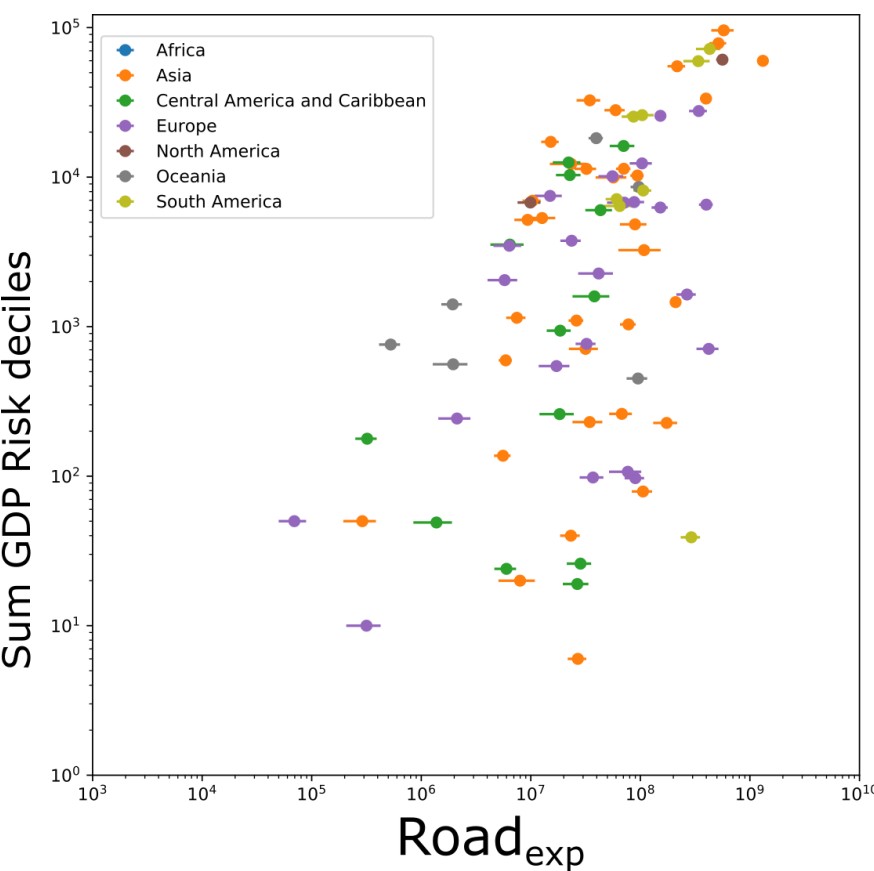

**Figure 6: Plotting the exposure of roads (in road-km Nowcasts / year) against the estimated GDP cost of landslide**
**impact estimated by Dilley et al. (2005).**


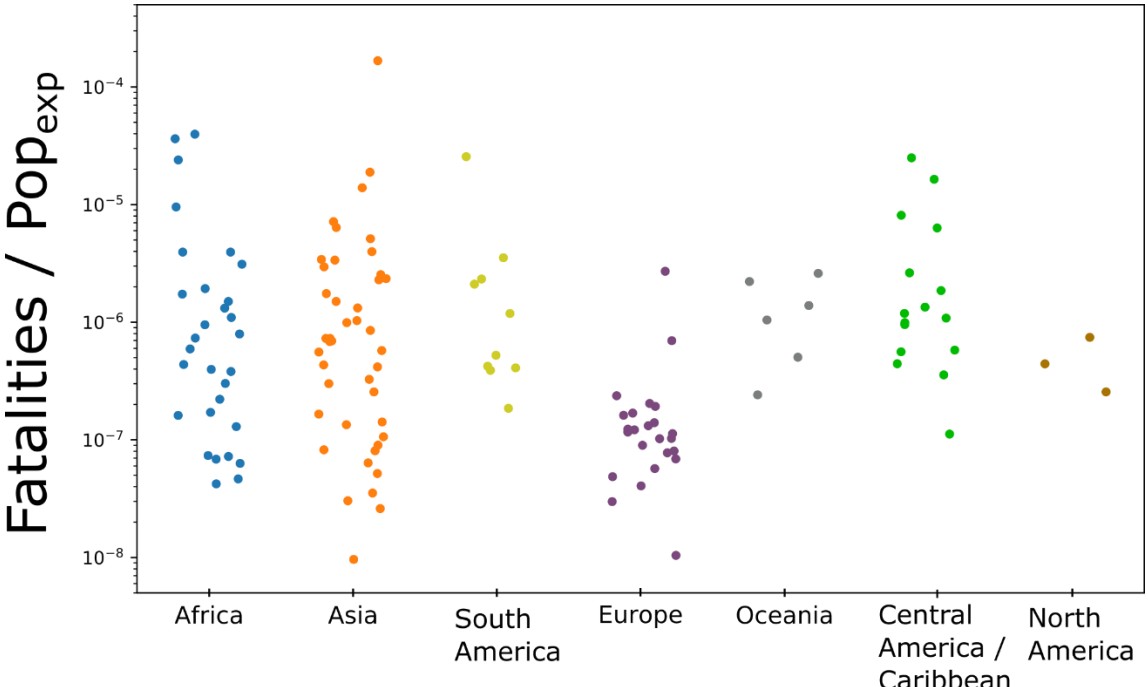

**Figure 7: Number of fatalities divided by Pop$_{exp}$, for each continent. The wide spread of values in Africa and Asia are likely a reflection of the diversity of nation-to-nation landslide vulnerability.**

445

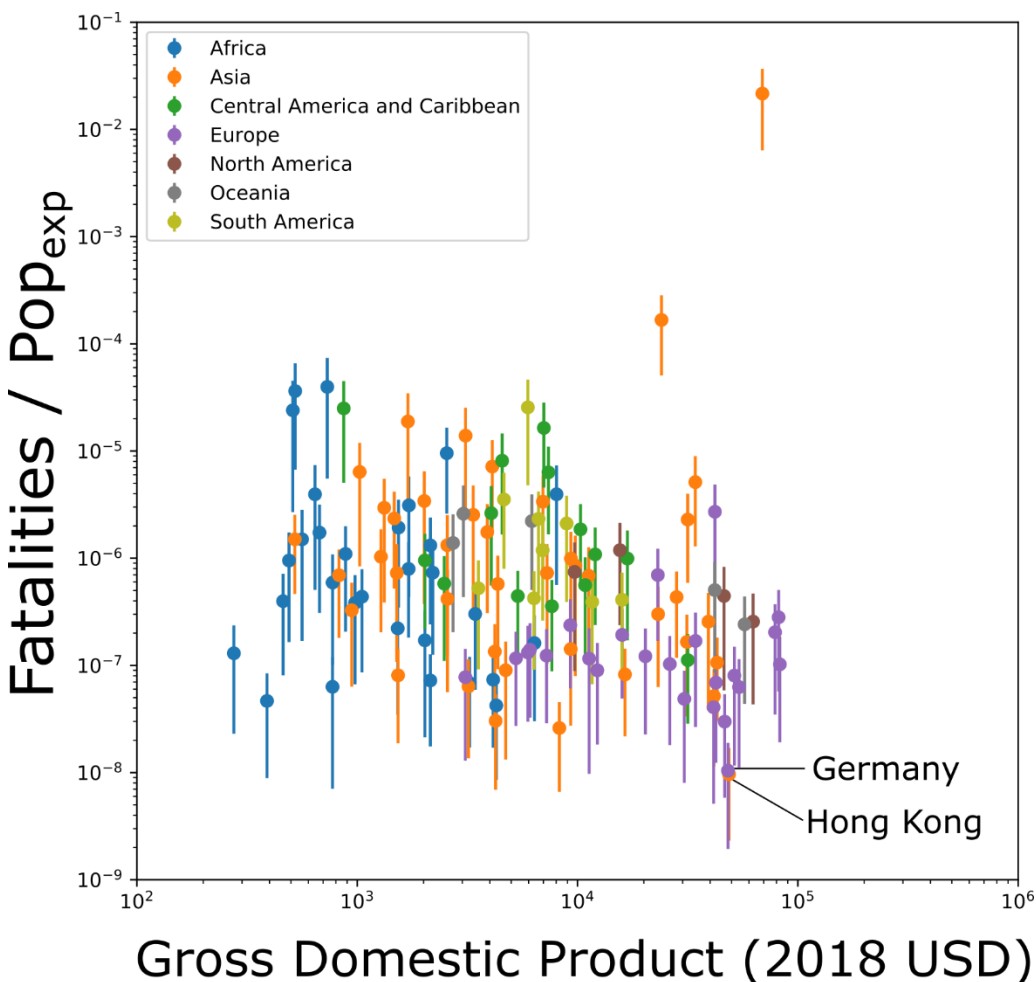

**Figure 8: Gross Domestic Product per capita (World Bank, 2018) compared with the number of landslide fatalities per unit exposure.**