# Peer review of "New Global Characterization of Landslide Exposure"

_Natural Hazards and Earth System Sciences, 2019_

## Referee Comment (RC1) · Anonymous Referee #1 · 13 Feb 2020

This is a really interesting paper, that I definitely think contributes something that doesn't currently exist: a homogeneous global measure of landslide exposure. The manuscript is well-written and there are not many problems with it that I can see. One general comment I would make is that the LHASA model is focused on modelling rapid landslides, and states that it may be less useful for estimating the occurrence of slow moving landslides. This is fine in terms of your estimation of population exposure, but you also consider damage to infrastructure which may be effected by slow-moving landslides and, so I think it should be stated somewhere that you are mostly considering rapid landslides to avoid confusion.

Specific comments

Line 55: From the wording of this sentence, it is not entirely clear whether it is the

output that is at 1 km resolution or the input data ( I assume it's both?)

Line 155: I agree - could this error be quantified better though, by comparing areas for which there is a relatively complete catalogue of landslides with the number of nowcasts received in that area?

Line 247 - which average? mean?

Figure 3 - Exactly what has been normalised by what is not really clear here e.g. in the text, it states that the population is normalised by country area but in the Figure it describes the Nowcasts as normalised but not the population

Line 283 - I agree that this would be a useful application of the study you have done here, but given the scatter in Figure 6, it seems that the uncertainty in y value for a given x must be very high.

Figures 7 & 8 - The conclusions you draw from Figure 7 could also be drawn from 8, so I am not convinced you need both

Line 344 - As the Petley dataset includes anthropogenic-induced landslides, could this account for some scatter in Figure 5?

Technical comments

Figure 1 & the top Figure in Figure 2 have very small numbers for the bin sizes that are quite hard to read

Line 158: estimate 'of' exposure, rather than 'for' exposure

Line 259 - Do you mean misrepresent instead of miss?

Line 281 'further highlights' not 'does further highlight'
* * *

---

## Referee Comment (RC2) · Anonymous Referee #2 · 28 May 2020

I enjoyed reading this paper and especially enjoyed looking at the maps and figures. It tackles an interesting and large problem of modeling relative landslide exposure worldwide. I think the topic the authors address is worthy of being published and the results will be of interest to many. However, the manuscript and results are not yet ready for publication. In particular, there is some potentially flawed logic and confusing methodological choices that need to be sorted out or clarified. I summarize the major points of confusion or issues that I feel need addressing below, and then provide line by line comments below.

1) The authors say they are modeling landslide exposure globally, and though they do not specify, from the context they give I assume they mean as it exists now or in general. However, I do not think that is actually what they are modeling. By using past

[Figure]

weather data, they are modeling what landslide exposure WAS averaged over the time period of the IMERG data. There were likely some extreme weather events in those 19 years that hit some countries and caused elevated Popexp values in this analysis that will not happen again in the next 19 years, there will be new extreme events that did not happen in the last 19 years, and also all bets are off with climate change.

2) The second major issue I struggled with was conceptualizing the physical meaning of the various metrics they use as proxies for exposure. What is the physical meaning of population exposure, road exposure and infrastructure exposure, as currently computed? What am I supposed to make of "nowcast density" or the inclusion of the vague concept of nowcasts in the units? Nowcasts are a fuzzy concept in themselves and then the authors convert it to another confusing metric, a rate of unknown timescale and a density over, I assume, time(?), it doesn't mean anything to me and is really hard to wrap one's mind around. Why not at least do something more tangible like number of days per year of elevated landslide hazard? Or alternatively, the percentage of the time that a given cell has an elevated "nowcast"? Nowcasts represent relative landslide hazard for 3 hour time periods, so either of those seem easy to compute and would make a lot more sense. It would give the units with some physical meaning, albeit still somewhat vague: e.g., people-hours/year/km2 exposed to elevated landslide hazard or percentage of the time landslide hazard is elevated/person/km2. Either would be much clearer in my opinion than the bizarre units and metrics currently used and also easier for other people to use or compare against in future studies.

3) The authors seem to conflate their modeled results with observations/ground truth and also conflate their modeled proxy for population exposure with actual population exposure. The authors need to always be clear that they are presenting modeled results and proxies, NOT data or observations. I have noted some of these instances in the line by line comments.

4) The writing is sometimes hard to follow. There are many run-on sentences and some confusing and/or convoluted logic, especially in the abstract and introductory sections.

I have pointed some of these out in the line by line comments.

5) Critical details are missing about the LHASA model updating that was done, the validation of the updated model (against, presumably, the inventories that were previously discussed as being biased?) as well as the uncertainty estimation.

6) One point needs clarifying: the authors state that they normalized the areas by squared decimal degrees rather than km2 for country-wide statistics. Do they mean they used the latitude and longitude grid? If so, that is going to skew the normalization pretty dramatically for countries away from the equator. That normalization needs to be done in units that preserve area for the proxy to be globally consistent and comparable from country to country.

Line by line:

Abstract: The logic of the abstract does not make sense to me. It introduces the problem of inventories being biased away from areas where human settlement or infrastructure are, but then it says in order to address this limitation, they are going to model global exposure to landslide hazard. . .but that isn't addressing the problem they raised in the previous sentence, they raised the problem of landslides in remote areas far from humans. It's also unclear what gaps in the inventory-based estimates (what estimates/estimates of what?) they are filling in. Then on line 17, they say they compare levels of landslide hazard "mitigation" between countries but they don't look at mitigation at all in the paper. Overall, the abstract is really confusing and could probably be rewritten.

L36 – though it is stated later, since the authors mentioned "near-real-time" here and that has many different meanings, I would suggest moving the info about how often it is updated and with what delay here.

L39 – why "near global"? what is missing?

L44-46 should be in the abstract

L49 – I would just delete the part of this sentence after the semicolon, no one should be using a global map at 1 km resolution for evaluating landslide hazard except as a way of looking at overall trends or patterns on a global/continental scale.

L50 – what pixel resolution? And I didn't think LHASA modeled exposure, does it? Isn't that what this paper is about?

L51 – it does not provide a "clear picture", that phrasing implies that it is reality. It is a model. An estimate. The authors should be clear about that here and throughout.

L54 – I find the concept of an "average rate of hazard Nowcast' to be very convoluted. See my earlier comment about a suggested alternative.

L57 – what is a "landslide climatology"?

L69 – what are these thresholds based on? Also, is it the ARI that is modeled based on the susceptibility layer or the threshold that the ARI must exceed? Please clarify.

L74 – Forest loss since when?

L75-79 – I found this description very hard to follow, can it be clarified?

L79-80 – More info needed on the methodology (generally, what is it based on/how does it work/how is it validated) even if the readers can still look to the original paper for the nitty gritty details.

L89-90 – How is landslide activity anticipated? Need more details here about how these thresholds are chosen since that ultimately controls all the results of this paper. . .

L92 – I thought there were different levels of "nowcasts", how are those dealt with in this averaging?

L108-109 – They happen to line up as well? Didn't you have to resample one to the other?

L117 – GRIP acronym needs to be defined above before its used.

L124-125 – Don't some of these road network datasets often include small "roads" like footpaths and farm roads? It seems like some levels might be worth excluding since they are not likely mapped consistently across the world whereas higher levels would be more likely to be consistent.

L140 – I am very confused about how one multiplies infrastructure by a nowcast density. . .

L149 – Need to include some more info about how the model is trained in the first place in order to understand this part about assessing errors.

L175 – Applying what? What is "this"?

L176-178 – This sentence is confusing, can it be clarified?

L180-182 – I'm not following the logic here, if the infrastructure is not completely mapped in a country, then the normalized metric will be just as wrong as the raw numbers, wouldn't it? The ratio would be off.

L195 – The results of this study are not observations. They are modeled proxies. Please rephrase.

L197 – Given the wonky units of popexp, population exposure annually is not what is modeled and shown in Figure 1, but a proxy for population exposure.

L229 – Impact is not the right word here. The authors are not modeling impact at all, they are modeling exposure, and also, the wording here implies that their results represent impacts that actually occurred, but they do not.

L240 – "modeled" population exposure

L256-258 – This point would be better made if Figure 4 showed the relative world map and the overall map side by side.

L259-260 – Don't we have the same problem if a given small country did not happen

to be hit by an extraordinary precipitation event in the 19-year IMERG dataset but was next year?

Figure 2C – it is hard to see the colors against the black background in this part of the figure.

Figure 5 – are the fatalities by country the total number ever, or per year, or ?

Figure 7 – what are the x-axis offsets within each continent? Just random for visibility?

Figure 8 – Here and in other similar plots, it might be useful to label some more key countries, especially the outliers.

―――――――――――――――――――――――

---

## Author Comment (AC1) · 9 Jul 2020

Response to Reviews for Emberson et al. 2020 NHESS-434

Dear Editor,

We greatly appreciate that you coordinated two thoughtful and helpful reviews for our study. We are glad that the reviewers seem to see the originality and value of our study. Both reviewers raise important points that we are happy to acknowledge to improve our study. We feel that we are able to effectively respond to each of the reviewer comments in a way that answers their concerns and therefore enhances the study. Below, we discuss the comments from both of the reviewers and explain our plan to adjust the study to incorporate their points. For most of the comments, we have already made changes, which are detailed for the relevant comments. There are other comments where we are still working to implement the changes (including in some cases reanalysis). In these cases we describe the approach that we are taking to address those changes. We look forward to your editorial consideration of our response.

Yours sincerely,

Robert Emberson, on behalf of all authors.

R1:

This is a really interesting paper, that I definitely think contributes something that doesn't currently exist: a homogeneous global measure of landslide exposure. The manuscript is well-written and there are not many problems with it that I can see.

We thank the reviewer for their supportive words, and we are glad they have identified the originality of our study as a strong point. We feel that the comments from this reviewer have allowed us to significantly improve our study.

One general comment I would make is that the LHASA model is focused on modelling rapid landslides, and states that it may be less useful for estimating the occurrence of slow moving landslides. This is fine in terms of your estimation of population exposure, but you also consider damage to infrastructure which may be effected by slow-moving landslides and, so I think it should be stated somewhere that you are mostly considering rapid landslides to avoid confusion.

This is an excellent point, and we appreciate the clarification. We have added the word 'rapid' in several places to clarify the type of failure we are studying, and have additionally added text to the end of the discussion section to address this:

*"In addition, the LHASA model only models rapid landslide failures in natural settings. This means it does not capture landslides resulting from anthropogenic influence or slow-moving landslide events, which lead to a significant number of fatalities every year (Petley, 2012). Constraining exposure to this kind of failure is another important subject for future studies."*

Specific comments

Line 55: From the wording of this sentence, it is not entirely clear whether it is the output that is at 1 km resolution or the input data (I assume it's both?)

We have rephrased this sentence to explain that the model outputs are at 30 arc-second resolution (approx. 1km at the equator).

Line 155: I agree - could this error be quantified better though, by comparing areas for which there is a relatively complete catalogue of landslides with the number of nowcasts received in that area?

This is a good point, but does raise some significant questions. There are inventories that are spatially complete mapping of landslides for given rainfall events, but multi-temporal mapping of landslides over even moderate time intervals is significantly less common. In order to effectively answer the question posed by the reviewer here, we need a multi-temporal inventory with year-to-year changes in landslide location mapped at relatively fine resolution. While such mapping has been conducted for some small portions of the earthquake-affected areas such as Nepal after the 2015 earthquake, the LHASA model is not designed to factor in post-earthquake changes in landslide susceptibility, meaning those inventories are not a fair test case. We are unaware of multi-temporal inventories mapped at the required temporal sampling (close to 1 year or lower, to maximize the comparison with short term nowcast outputs) in published literature. Italy has undertaken a tremendous effort to create multi-temporal inventories in many regions; however, these are typically done at 5 year intervals and is not fully comprehensive. The lead author (Emberson) has worked with multi-temporal inventories in the Western Southern Alps of New Zealand, but the mapping frequency is approximately decadal, rather than yearly, which we do not feel is appropriate for comparison with LHASA outputs. We stress that if the reviewer or other reader is aware of such an inventory in published literature, we would be delighted to hear about it, as it would represent an exceptional test of our model outputs, which the reviewer correctly points out would be a valuable step forward.

Line 247 - which average? mean?

Good catch! We have amended to clarify that this is the mean.

Figure 3 - Exactly what has been normalised by what is not really clear here e.g. in the text, it states that the population is normalised by country area but in the Figure it describes the Nowcasts as normalised but not the population

Again, thank you for catching this. We have amended the text to correct this; the figure shows Pop_exp vs area-normalised nowcasts. Pop_exp is not normalized by area. This should clear up the discrepancy.

Line 283 - I agree that this would be a useful application of the study you have done here, but given the scatter in Figure 6, it seems that the uncertainty in y value for a given x must be very high.

We agree that this relationship is at present only loosely defined. We do not attempt to use it in an interpretive way here. We have added the following text to emphasise the point of the reviewer:

*"However, the degree of scatter evident in Figure 6 suggests that further data is required to explicitly define such a relationship, and error margins may be large"*

Figures 7 & 8 - The conclusions you draw from Figure 7 could also be drawn from 8, so I am not convinced you need both

While we acknowledge that the same data is shown on the y-axis for both figures, we feel that Figure 7 allows for clearer comparison of countries in each continent, which are rather overlapping in Figure 8. Unless directed by the editor, we are inclined to keep both figures to ensure clarity.

Line 344 - As the Petley dataset includes anthropogenic-induced landslides, could this account for some scatter in Figure 5?

As discussed above in our response to the main comment, we have added text here to incorporate this point. The reviewer is absolutely correct and we appreciate them bringing it up here! New text:

*"In addition, the LHASA model only models rapid landslide failures in natural settings. This means it does not capture landslides resulting from anthropogenic influence or slow-moving landslide events, which lead to a significant number of fatalities every year (Petley, 2012). Constraining exposure to this kind of failure is another important subject for future studies."*

Technical comments

Figure 1 & the top Figure in Figure 2 have very small numbers for the bin sizes that are quite hard to read

Increased font size for bin sizes; thanks for pointing this out!

Line 158: estimate 'of' exposure, rather than 'for' exposure

Changed to 'estimate of'.

Line 259 - Do you mean misrepresent instead of miss?

Changed to 'misrepresent'.

Line 281 'further highlights' not 'does further highlight'

Changed to 'further highlights'.

Reviewer 2:

I enjoyed reading this paper and especially enjoyed looking at the maps and figures. It tackles an interesting and large problem of modeling relative landslide exposure worldwide. I think the topic the authors address is worthy of being published and the results will be of interest to many. However, the manuscript and results are not yet ready for publication. In particular, there is some potentially flawed logic and confusing methodological choices that need to be sorted out or clarified. I summarize the major points of confusion or issues that I feel need addressing below, and then provide line by line comments below.

We thank the reviewer for taking the time to provide an extremely thorough and thoughtful review. We certainly appreciate the support for the study concept, and acknowledge that there remains room for improvement. All of the comments are helpful and have allowed us to improve our study. We have provided individual responses to each of the comments below.

1) The authors say they are modeling landslide exposure globally, and though they do not specify, from the context they give I assume they mean as it exists now or in general. However, I do not think that is actually what they are modeling. By using past weather data, they are modeling what landslide exposure WAS averaged over the time period of the IMERG data. There were likely some extreme weather events in those 19 years that hit some countries and caused elevated Popexp values in this analysis that will not happen again in the next 19 years, there will be new extreme events that did not happen in the last 19 years, and also all bets are off with climate change.

The reviewer raises a good point here, and is absolutely correct that our model outputs are inappropriate for assessment of exposure under future climatic changes. We have adjusted the text in several places to better stress this, as well as highlight the value of longer term historical data and future climate data to better quantify exposure.

End of introduction: *"While the model outputs are an approximation of exposure to hazard based on historical rainfall trends, we note that future exposure patterns could be explored with the use of rainfall projections for future climate scenarios."*

In discussion section: *"In addition, since the nowcast-based estimates of hazard are based on historical rainfall data, they do not provide effective prediction of future exposure to hazard. This is particularly important given the potential for climate change to affect rainfall-driven hazards (Kleinen & Petschel-Held 2007). Our model estimates of exposure would also fail to capture rainfall driven exposure to landslide hazards in periods outside of the IMERG v06B record (pre 2001), including major rainfall-driven landslide events resulting from the 1998 El Nino event (Coe et al. 2004, Ngecu & Mathu 1999). We stress that the model outputs are representative of the historical period under analysis, rather than strictly speaking a long-term average."*

2) The second major issue I struggled with was conceptualizing the physical meaning of the various metrics they use as proxies for exposure. What is the physical meaning of population exposure, road exposure and infrastructure exposure, as currently computed? What am I supposed to make of "nowcast density" or the inclusion of the vague concept of nowcasts in the units? Nowcasts are a fuzzy concept in themselves and then the authors convert it to another confusing metric, a rate of unknown timescale and a density over, I assume, time(?), it doesn't mean anything to me and is really hard to wrap one's mind around. Why not at least do something more tangible like number of days per year of elevated landslide hazard? Or alternatively, the percentage of the time that a given cell has an elevated "nowcast"? Nowcasts represent relative landslide hazard for 3 hour time periods, so either of those seem easy to compute and would make a lot more sense. It would give the units with some physical meaning, albeit still somewhat vague: e.g., people-hours/year/km2 exposed to elevated landslide hazard or percentage of the time landslide hazard is elevated/person/km2. Either would be much clearer in my opinion than the bizarre units and metrics currently used and also easier for other people to use or compare against in future studies.

We are grateful to the reviewer for raising this point, as it has helped us improve our message and clarify the terminology. We have extensively revised several sections of the text to explain the physical meaning of a nowcast, and have propagated that understanding throughout the rest of the text. We have not removed all mention of nowcasts, since to do so might lead the reader to believe that we are using a different hazard proxy than what we derive from LHASA. However, we feel that in explaining the physical meaning and clarifying the usage throughout the text, we have reduced the ambiguity highlighted by the reviewer here.

Changes to text:
Section 2.1: *"For the purposes of our study, we use the daily nowcast output. The physical meaning of one nowcast is 24 hours of elevated landslide hazard for a 30 arc-second dimension pixel."*

Section 2.2: Revision of table 1 to explain units of exposure more accurately; we replace the term 'nowcast' with 'days exposed to landslide hazard'. This means that Pop_exp is (Days exposed to landslide hazard) x (persons) per year per km^2.

3) The authors seem to conflate their modeled results with observations/ground truth and also conflate their modeled proxy for population exposure with actual population exposure. The authors need to always be clear that they are presenting modeled results and proxies, NOT data or observations. I have noted some of these instances in the line by line comments.

This is a very important clarification, and one we appreciate from the reviewer. It is absolutely correct that these are model outputs, rather than ground-truth. We have revised the document to clearly state this difference, including by adding 'model outputs' where possible to drive this home. We have also addressed the individual line-by-line edits below.

4) The writing is sometimes hard to follow. There are many run-on sentences and some confusing and/or convoluted logic, especially in the abstract and introductory sections. I have pointed some of these out in the line by line comments.

We thank the reviewer for taking the time to highlight these places. Clarity is obviously key here, so all the input is appreciated! We have addressed the individual comments below.

5) Critical details are missing about the LHASA model updating that was done, the validation of the updated model (against, presumably, the inventories that were previously discussed as being biased?) as well as the uncertainty estimation.

Again, this is an important point and we are grateful to the reviewer for highlighting it. We did not make significant changes to the methodology of Stanley & Kirschbaum (2017) when updating our susceptibility map, but it is still necessary to provide more details as well as provide quantification of validation. As such, we have added text to the methodology section to explain the model in more detail. We are also calculating the ROC-AUC values for the updated susceptibility map with the Global Landslide Catalog as validation, and although that has not been completed in time for the final author comment part of this review process, we will be adding those values to the revised manuscript (i.e., this is a planned action to address this comment; in the new text below, the X values will be replaced with results, and we will tabulate the ROC-AUC results).

New text: *"This fuzzy overlay model uses heuristic weighting of the input variables, defined by Stanley & Kirschbaum (2017). We do not adjust the weights attached to the variables in the study here. We assess the accuracy of the new susceptibility map in the same fashion as in the study of Stanley & Kirschbaum (2017), by using the NASA Global Landslide Catalog locations to test the ROC-AUC values. The updated model values are: [to be added once finalized]. For the purposes of our analysis, we follow Stanley and Kirschbaum (2017) and use susceptibility values of greater than 0.49 (on a 0-1 scale) as a threshold for nowcasts to be generated if rainfall exceeds the historical 95th percentile. [X] proportion of landslides in the GLC occur in areas above this threshold. For the purposes of this study, we combine moderate and high 'nowcasts' together to provide a proxy for hazard that captures the bulk of landslide activity."*

6) One point needs clarifying: the authors state that they normalized the areas by squared decimal degrees rather than km2 for country-wide statistics. Do they mean they used the latitude and longitude grid? If so, that is going to skew the normalization pretty dramatically for countries away from the equator. That normalization needs to be done in units that preserve area for the proxy to be globally consistent and comparable from country to country.

We thank the reviewer for bringing this up as it allows us to fix a key issue with the initial manuscript. While we had initially described results as 'approximately 1km resolution', this is strictly speaking not true as all input data is either directly sourced at 30 arc-second resolution or (in the case of roads and infrastructure) calculated on a grid at that resolution. As such, normalization by dividing by area in

square decimal degrees does preserve the area for the proxy. We have adjusted the text to explain everywhere that this is at 30 arc-second resolution, rather than 1km resolution.

Line by line:

Abstract: The logic of the abstract does not make sense to me. It introduces the problem of inventories being biased away from areas where human settlement or infrastructure are, but then it says in order to address this limitation, they are going to model global exposure to landslide hazard. . .but that isn't addressing the problem they raised in the previous sentence, they raised the problem of landslides in remote areas far from humans. It's also unclear what gaps in the inventory-based estimates (what estimates/estimates of what?) they are filling in. Then on line 17, they say they compare levels of landslide hazard "mitigation" between countries but they don't look at mitigation at all in the paper. Overall, the abstract is really confusing and could probably be rewritten.

The reviewer raises a good point here, since there is indeed some logical inconsistency in the abstract. We have revised it extensively to fix this:

*"Landslides triggered by intense rainfall are hazards that impact people and infrastructure across the world, but comprehensively quantifying exposure to these hazards remains challenging. Unlike earthquakes or flooding which cover large areas, landslides occur only in highly susceptible parts of a landscape affected by intense rainfall, which may not intersect human settlement or infrastructure. Existing datasets of landslides around the world generally include only those reported to have caused impacts, leading to significant biases toward areas with higher reporting capacity, limiting how our understanding of exposure to landslides in developing countries. In this study, we use an alternative approach to estimate exposure to landslides in a homogenous fashion. We have combined a global landslide hazard proxy derived from satellite data with open-source datasets on population, roads and infrastructure to consistently estimate exposure to rapid landslide hazards around the globe. These exposure models compare favourably with existing datasets of rainfall-triggered landslide fatalities, while filling in major gaps in inventory-based estimates in parts of the world with lower reporting capacity. Our findings provide a global estimate of exposure to landslides from 2001-2019 that we suggest may benefit disaster mitigation professionals."*

L36 – though it is stated later, since the authors mentioned "near-real-time" here and that has many different meanings, I would suggest moving the info about how often it is updated and with what delay here.

We have updated the paragraph to more explicitly define the previously published LHASA model, explaining the 50N-50S spatial extent and explaining the 4-hour latency of results.

L39 – why "near global"? what is missing?

See prior comment – added in text to explain that the previously published LHASA outputs are 50N-50S.

L44-46 should be in the abstract

We agree with the reviewer that this sentence was not appropriate for this position, so we have removed it and merged the text with the prior paragraph. The abstract has been revised to accommodate other comments from this reviewer, so we have not moved this sentence there.

L49 – I would just delete the part of this sentence after the semicolon, no one should be using a global map at 1 km resolution for evaluating landslide hazard except as a way of looking at overall trends or patterns on a global/continental scale.

Thanks for pointing this out – the text is definitely unnecessary and inappropriate. We've deleted it.

L50 – what pixel resolution? And I didn't think LHASA modeled exposure, does it? Isn't that what this paper is about?

Thank you for flagging this confusion; we have rephrased this accordingly:
*"Our exposure model outputs derived from the LHASA model provide an estimate of exposure seasonality at 30 arc second resolution across the globe"*

L51 – it does not provide a "clear picture", that phrasing implies that it is reality. It is a model. An estimate. The authors should be clear about that here and throughout.

We have rephrased this to "to provide a more spatially consistent picture of the impacts associated with landslides"

L54 – I find the concept of an "average rate of hazard Nowcast' to be very convoluted. See my earlier comment about a suggested alternative.

Removed this sentence as part of rephrasing the opening paragraph to the methodology section. We have revised this to emphasise that we draw the hazard estimates from the updated LHASA model, as well as clarify the resolution (30 arc-seconds):

*"To estimate exposure to landslide hazard, we must first derive the estimates of hazard itself. For this study, we have utilised the outputs of an updated version of the LHASA model as an approximation for hazard, which we can then combine with openly available datasets of infrastructure at a 30 arc-second resolution across the world."*

L57 – what is a "landslide climatology"?

Rephrased this sentence: *"These maps of exposure, both annually and estimated for each month to analyse seasonal variability"*

L69 – what are these thresholds based on? Also, is it the ARI that is modeled based on the susceptibility layer or the threshold that the ARI must exceed? Please clarify.

Added text to clarify: *"If the ARI value exceeds a threshold (historical 95th percentile for rainfall)"*

L74 – Forest loss since when?

Thanks for flagging! We have amended the text to explain that this is forest loss since the year 2000.

L75-79 – I found this description very hard to follow, can it be clarified?

We have made adjustments to this section to hopefully improve clarity.

L79-80 – More info needed on the methodology (generally, what is it based on/how does it work/how is it validated) even if the readers can still look to the original paper for the nitty gritty details.

We appreciate the need for more detail on the methodology. We have added more text to explain the susceptibility model detail, as well as to address major comment #5, above. Please note that as in comment #5, above, we are still in the process of calculating the updated ROC-AUC values, so while these will be included in the revised version of the manuscript, we have not been able to include them in the author response part of this review process. New text:

*"This fuzzy overlay model uses heuristic weighting of the input variables, defined by Stanley & Kirschbaum (2017). We do not adjust the weights attached to the variables in the study here. We assess the accuracy of the new susceptibility map in the same fashion as in the study of Stanley & Kirschbaum (2017), by using the NASA Global Landslide Catalog locations to test the ROC-AUC values. The updated model values are: [to be added]. For the purposes of our analysis, we follow Stanley and Kirschbaum (2017) and use susceptibility values of greater than 0.49 (on a 0-1 scale) as a threshold for nowcasts to be generated if rainfall exceeds the historical 95th percentile. [X] proportion of landslides in the GLC occur in areas above this threshold. For the purposes of this study, we combine moderate and high 'nowcasts' together to provide a proxy for hazard that captures the bulk of landslide activity."*

L89-90 – How is landslide activity anticipated? Need more details here about how these thresholds are chosen since that ultimately controls all the results of this paper. . .

We have revised this section, so that it is hopefully now more clear:

"The LHASA model generates a hazard 'nowcast' if rainfall exceeds the historical 95th percentile. Since the updated model uses IMERG v06B rather than TMPA, we have therefore re-calculated the historical 95th percentiles of a 7-day weighted rainfall accumulation. This provides a global 95th percentile map; if ARI values exceed this threshold, a hazard nowcast is issued."

L92 – I thought there were different levels of "nowcasts", how are those dealt with in this averaging?

We have addressed this in the updated discussion of the LHASA methodology, as detailed in the response to comment #5, above. The new text is as follows:

*"For the purposes of this study, we combine moderate and high 'nowcasts' together to provide a proxy for hazard that captures the bulk of landslide activity."*

L108-109 – They happen to line up as well? Didn't you have to resample one to the other?

As discussed elsewhere, this is actually one of the advantages of using this data. The GPW v4 data and the LHASA outputs are derived at identical resolutions – 30 arc-seconds. This means no resampling is necessary. Ultimately, resampling the population dataset to get at a 1km dataset ultimately requires interpolation of data and the assumption that to do so does not inhibit the data quality. As such, we feel that the current lat-long (rather than 1km scale exactly) model is more scientifically defensible. We have amended the text in various places to explain this; see response to main comment #6, above, for full details.

L117 – GRIP acronym needs to be defined above before its used.

Added acronym definition after first use (Global Roads Inventory Project)

L124-125 – Don't some of these road network datasets often include small "roads" like footpaths and farm roads? It seems like some levels might be worth excluding since they are not likely mapped consistently across the world whereas higher levels would be more likely to be consistent.

Thanks for flagging – the GRIP dataset does not include this kind of road (essentially it's tarmac roads or larger) so issue is of lower concern. We have added a sentence to clarify:

*"This dataset does not include footpaths or unpaved roads, for which mapping may be significantly more spatially inconsistent."*

L140 – I am very confused about how one multiplies infrastructure by a nowcast density. . .

We have revised this sentence to hopefully alleviate the confusion! New text:

*"To combine the roads datasets and OSM-derived critical infrastructure with the hazard outputs, we have multiplied the raster map of infrastructure or road density by the Nowcast density raster (i.e. raster showing total days exposed to landslide hazard)"*

L149 – Need to include some more info about how the model is trained in the first place in order to understand this part about assessing errors.

We feel that this has now been appropriately addressed by changes made to address earlier comments; see in particular the response to comment #5, above.

L175 – Applying what? What is "this"?

L176-178 – This sentence is confusing, can it be clarified?

We address both of the two comments above in one – we've changed this entire sentence to improve clarity:

*"The OSM completeness estimates are calculated at a national level, and it is therefore not clear how to apply them to the 30 arc-second pixels in our study, and as such we do not attempt to correct our global maps. However, to effectively normalise the exposure data at a country level, we provide the completeness measure derived from Barrington-Leigh and Millard-Ball (2017) in Supplementary Table 1"*

L180-182 – I'm not following the logic here, if the infrastructure is not completely mapped in a country, then the normalized metric will be just as wrong as the raw numbers, wouldn't it? The ratio would be off.

We are unsure of what the reviewer means here; the infrastructure exposure in each country in the supplemental figures is shown as 'exposed' elements divided by total elements. In other words, it's the proportion of total mapped elements exposed to landslide. So comparison between countries focuses on the exposed fraction. If mapping is incomplete, this does not necessarily mean the exposed fraction will be affected, if the completeness is the same across the country. Since we only have completeness estimates at a national level, we can't tell if this is true or not. Ultimately, exposed fraction gives a normalized sense of 'how much in each country is exposed' which is a more useful comparison that looking at total exposed elements. As such, we feel that the original statement is justified, although we would be happy to discuss.

L195 – The results of this study are not observations. They are modeled proxies. Please rephrase.

Changed to 'modeled estimates'.

L197 – Given the wonky units of popexp, population exposure annually is not what is modeled and shown in Figure 1, but a proxy for population exposure.

Changed to "Figure 1 shows the modeled estimates of population exposure"

L229 – Impact is not the right word here. The authors are not modeling impact at all, they are modeling exposure, and also, the wording here implies that their results represent impacts that actually occurred, but they do not.

Thanks for this – you're absolutely correct. Changed the text to: "the exposure of linear infrastructure to landslides is more widespread".

L240 – "modeled" population exposure

Added 'modeled'

L256-258 – This point would be better made if Figure 4 showed the relative world map and the overall map side by side.

Good point. We will add this to the revised version.

L259-260 – Don't we have the same problem if a given small country did not happen to be hit by an extraordinary precipitation event in the 19-year IMERG dataset but was next year?

This is an important point. The key factor is that all rainfall above the 95$^{th}$ percentile will lead to the same nowcast output – so the model isn't massively biased by e.g. 100-year return period rainfall event, but at the same time it is relatively insensitive to those enormous rainfall events. We have added the following sentence to explain this:

*"At the same time, the LHASA-based model outputs are relatively insensitive to extreme rainfall events (100-year return period, for example), since all rainfall values above the 95th historical percentile will lead to the same nowcast hazard output."*

Figure 2C – it is hard to see the colors against the black background in this part of the figure.

Thanks for raising this. We will revise this figure to be clearer.

Figure 5 – are the fatalities by country the total number ever, or per year, or ?

Changed text to explain that this is total fatalities: "*Figure 5: Showing the exposure of population (in person-Nowcasts/year) against the number of total fatalities recorded in the dataset of Froude and Petley 2018*"

Figure 7 – what are the x-axis offsets within each continent? Just random for visibility?

That's correct, they're to avoid overlap. We've added text to the caption to explain this:

*"Offsets in the x-axis are for visual distinction between points to avoid overlap."*

Figure 8 – Here and in other similar plots, it might be useful to label some more key countries, especially the outliers

This is a good point. We have added labels to Qatar and Bahrain, since they are explicitly named in the text. We do not seek to draw attention to other countries, as we do not feel it is our place to 'name and shame' in any respect.

---

## Author Response (AR2)

Response to Editor and Reviewers - Minor Revisions, Emberson et al. NHESS 434

Dear Editor,

Thank you for coordinating a useful set of follow-up reviews for our revised study. We found them both helpful and we feel that we have addressed each of the comments.
To respond specifically to the concern raised in your (the editor) response, since the time allowed to respond to the reviewers (the NHESS response phase) was not sufficient to re-run the entire analysis to re-generate the ROC-AUC scores, in our response to the reviewers we
explained that we were doing that analysis, but did not have the numbers calculated. However, since we had more time before we had to submit the revised manuscript, we calculated the final numbers and added them into the revised text. As such, the response to the reviewers represents a 'plan' for what we are doing. I hope this is now clear.
Below, we have included the reviewer comments from this round of reviews and our responses
to each point (essentially all minor technical corrections). The marked up version of the manuscript is below.

We are seeking internal approval to upload all relevant raw data (supplementary material already includes all necessary results, but we would like to include raw raster data) so we
wanted to check – is it possible to amend supplementary material after publication once we have uploaded raster and shapefile data to a repository?
We look forward to hearing back in due course.

Sincerely, on behalf of all authors,
Robert Emberson

Reviewer #1 Comments:
Line 58 - the sentence starts with a full stop
Fixed
Line 115 - Missing close bracket
Fixed
Table 1 - Why is there a mixture of . and x in the population exposure units?
Replaced 'x' with '.'
Line 214 - There is an unnecessary comma before "respectively"
Removed comma
Line 304 - Replace 'is' with 'are'
Changed
Figure 4 - There are faint boxes around the keys on the left hand side of these panels, but the upper box has a side missing and the lower one has "Popexp /population" over the top. This is a
bit untidy, I think these boxes should be removed
Fixed, removed boxes
- Why are some countries (e.g. Somalia) coloured black on the popexp/population map. Were there no population data available? Or is the value of popexp/population outside the colour scale for these countries? Please specify this in the figure caption.

You are correct, we lack any population data for some countries from the World Bank dataset. Explained this in the caption.
Figure 4, caption - "exp" is not written as a subscript in Popexp, which it is elsewhere in the manuscript and in the Figure itself.
Fixed

Figures 3,5,6,8 - the font size of the axis labels seems to vary throughout these, the paper would look neater if they were all the same. In particular it looks like Popexp is written in a larger font where it appears. (I apologise if this is not the case and my eyes are playing tricks on me).
Fixed this. It was partly that the figures are displaying at different sizes in word, so we will work with the editorial staff to ensure the figures are displayed at the appropriate size. We have also
adjusted the font sizes to be equal.

Comments from Reviewer #2:
I think the authors did a great job responding to the reviews and the paper is nearly ready for publication with a few minor corrections that I listed below:
Thank you to the reviewer for the support. We have addressed each of the comments, which have proven helpful to finalise our study.

I suggest the authors include the raster files and shapefiles for the layers shown in the figures in the supplement (or another data archiving site), or at least releasing a kmz file with the paper.
That will make it a lot more usable by others.
The rasters and shapefiles will be uploaded with the final manuscript.
Figures: The figures are referenced out of order in the text, Fig 5 is presented in text before 3 and 4. Some discussion about some figures (Fig 4 in particular) occurs prior to the figure being presented in the text.
Moved figure 4 to above where it is first mentioned, removed mention of figs 5 & 6 prior to their introduction properly in discussion.
Many of the tick labels on the figures are far too small and need to be increased. For the scatterplots (Figs 3, 5, 6, 7), I really wanted to know what countries were which when they were discussed in the text, so I suggest labeling select dots that are mentioned in the text and also
labeling any outliers (like the authors already did in Figure 8)
Increased size of tick labels for all figures, added labels for outliers.
Figure 2: This figure is still hard to see because the pixels are so small relative to the size of the map, I would suggest showing the same zoomed in area for all three (Switzerland is fine) but also modifying the color maps and/or using some pyramiding/resampling methods to better
visualize the plots at the scale of the map.
We have tried several versions of this figure but have been unable to significantly improve the clarity while preserving the information. Using pyramids does not seem to help, and moreover means some information is lost. We are happy to work with the editorial staff to ensure this is as clear as possible, but even when zoomed into Switzerland the small pixel aspect remains. We
prefer to retain the larger scale to illustrate the changes across a broad region, even if it has a somewhat pixelated nature.
Figure 4: This figure should be presented in the text in the results section, it is one of the main results of the paper so it can be bigger so it is easier to see. It also should be cleaned up by rounding the labels in the legend, making sure the legends don't overlap and possibly adding
arrows pointing to some regions of interest that are discussed in the text.
       Fixed legends. The size of the figure will depend on editorial specifications – we can provide a
       figure at any resolution, but it depends what space is available in the published version to
       illustrate it.
       Added text to results section to explain this figure.
Line by line (line numbers are from the tracked changes document)

       L414: may benefit -> may be useful to
       Changed accordingly
       L488: state what the official levels are here (low, moderate, high...), because later they are used
(moderate, high etc.) but it seems like those are just qualitative terms later in the text, but
       actually they are explicitly defined by LHASA.
       Added clarification text:
       "This susceptibility is divided into categories based on decreasing area of the world occupied by
       each increasing class: this classification scheme was designed so that each category was twice
as large as the next highest, e.g., the very low category contains roughly twice as many pixels
       as the low category."
       L492-494: How is ARI weighted?
       Added clarification text:
       "The weighting is an exponential weighting, with each day prior to the most recent multiplied by
$1/n2$, where n is the number of days prior to present. The exponent value of 2 was calculated by
       Kirschbaum and Stanley (2018) based on calibration at the locations of 949 landslides from the
       years 2007-2013."
       L521: What is definition of "historical" here? The time period of the satellite precip data? If so,
       that's not very historical because it doesn't go back very far. Or do the authors mean a longer
term precipitation record?
       This is based on the satellite precip data, yes. We have added text to clarify. Using satellite
       precip data to calculate percentiles avoids inconsistencies. While the reviewer is correct that this
       does not cover a full climatological timescale, we feel the term historical is still appropriate – this
       is, after all, historical data.
L602-604: Are these point locations? If so, is the density computation done assuming they are
       all of equal area?
       This is correct, they are counted as a single 'school' or 'hospital', not 'school area'. The density
       is in units of 'schools per 30x30 arc second cell' for example. Added clarifying text:
       "We count each node as a single point, providing a density estimate of 'nodes [school or
hospital, etc] per 30x30 arc second cell', where nodes are of the types defined."
       L650-657: Also clarify that this requires the assumption that the completeness is uniform across
       each country.
       Added sentence: "Note that this also assumes completeness is consistent within individual
       countries"
L729: What figure should the reader look at to follow this paragraph of the discussion?
       Added text to explain this refers to Figure 1
       L779: Move this to the results

Moved to results

L781: It might be interesting to add a figure of the US or Brazil broken down by country or other internal subdivision to illustrate this point and show the distribution.

This is a good point, but we prefer not to add this here. We are developing a fuller assessment of impacts by subdivision, and our preference is to preserve that as a separate, more comprehensive piece of work.

L842: Citation needed for the extensive mitigation efforts of Germany and Hong Kong if this is a known fact. Otherwise, clarify that this is speculation.

Clarified that this is speculative

Marked up text:

[revised manuscript text omitted]